# Is Drinking Alcohol Really Linked to Cardiovascular Health? Evidence from the Kardiovize 2030 Project

**DOI:** 10.3390/nu12092848

**Published:** 2020-09-17

**Authors:** Andrea Maugeri, Ota Hlinomaz, Antonella Agodi, Martina Barchitta, Sarka Kunzova, Hana Bauerova, Ondrej Sochor, Jose R. Medina-Inojosa, Francisco Lopez-Jimenez, Manlio Vinciguerra, Gorazd Bernard Stokin, Juan Pablo González-Rivas

**Affiliations:** 1International Clinical Research Center, St Anne’s University Hospital, 65691 Brno, Czech Republic; ota.hlinomaz@fnusa.cz (O.H.); sarka.kunzova@fnusa.cz (S.K.); hana.bauerova@fnusa.cz (H.B.); ondrej.sochor@fnusa.cz (O.S.); manlio.vinciguerra@fnusa.cz (M.V.); gorazd.stokin@fnusa.cz (G.B.S.); juanpgonzalezr@hotmail.com (J.P.G.-R.); 2Department of Medical and Surgical Sciences and Advanced Technologies “GF Ingrassia”, University of Catania, 95127 Catania, Italy; agodia@unict.it (A.A.); martina.barchitta@unict.it (M.B.); 3Division of Preventive Cardiology, Department of Cardiovascular Medicine, Mayo Clinic and Mayo Medical School, Rochester, MI 55905, USA; MedinaInojosa.Jose@mayo.edu (J.R.M.-I.); lopez@mayo.edu (F.L.-J.); 4Department of Global Health and Population, Harvard T.H. Chan School of Public Health, Boston, MA 02115, USA

**Keywords:** drinking habits, cardiovascular disease, nutritional epidemiology, cardiometabolic health, public health

## Abstract

Existing data have described benefits and drawbacks of alcohol consumption on cardiovascular diseases (CVD), but no research has evaluated its association with the cardiovascular health (CVH) score proposed by the American Heart Association. Here, we conducted a cross-sectional analysis on the Kardiovize cohort (Brno, Czech Republic), to investigate the relationship between alcohol consumption and CVH. We included 1773 subjects (aged 25–64 years; 44.2% men) with no history of CVD. We compared CVD risk factors, CVH metrics (i.e., BMI, healthy diet, physical activity level, smoking status, blood pressure, fasting glucose, and total cholesterol) and CVH score between and within several drinking categories. We found that the relationship between drinking habits and CVH was related to the amount of alcohol consumed, drinking patterns, and beverage choices. Heavy drinkers were more likely to smoke tobacco, and to report diastolic blood pressure, fasting glucose, triglycerides, and low-density lipoprotein (LDL)-cholesterol at higher level than non-drinkers. Among drinkers, however, people who exclusively drank wine exhibited better CVH than those who exclusively drank beer. Although our findings supported the hypothesis that drinking alcohol was related to the CVH in general, further prospective research is needed to understand whether the assessment of CVH should incorporate information on alcohol consumption.

## 1. Introduction

Alcohol consumption is an integral part of many cultural, religious, and social practices, and provides perceived enjoyment to many users. A report by the World Health Organization (WHO) showed that the worldwide yearly alcohol consumption per capita in adults rose from 5.5 L of pure alcohol in 2005 to 6.4 L in 2016, with the highest levels in Central and Eastern Europe [1]. The same report also highlighted that alcohol consumption contributes to many deaths and disability globally, with more than 3 million deaths (5.3% of all deaths) and 132.6 million disability-adjusted life years (DALYs) as a harmful result of excessive consumption [1]. In the same year, 0.6 million deaths attributable to alcohol consumption were due to cardiovascular diseases (CVD), nearly 20% of all alcohol-related deaths [1]. As stated by O’Keefe and colleagues, alcohol is like a double-edged sword, which can cut so deeply in either direction—toxic or beneficial depending on how it is used [2]. Indeed, the supposed benefits from drinking are dependent on quantity and pattern of alcohol consumption [2]. Most studies have reported a J-shaped association with total and CVD mortality, whereby light-to-moderate drinkers had less risk than abstainers, and heavy drinkers were at the highest risk [2]. With respect to cardio-metabolic diseases, regular light-to-moderate alcohol consumption has been associated with lower incidence of coronary artery disease, diabetes mellitus, congestive heart failure, and stroke [3]. In contrast, excessive alcohol consumption has been associated with hypertension, atrial fibrillation, and non-ischemic dilated cardiomyopathy in a dose-dependent fashion [3]. Among the different types of beverages, red wine is generally associated with the best cardiovascular benefits [4,5,6,7], probably due to its unique array of non-alcoholic components (i.e., bioflavonoids with antioxidant, antiplatelet, and antiendothelin-1 effects) [8]. However, some lines of evidence suggest that the amount and pattern of alcohol consumption are more important than specific alcoholic preferences [9]. Indeed, alcohol itself, rather than specific components of drinks, might be the major factor in conferring health benefits [10,11], with some studies showing equal protection from wine, beer, or spirits [3]. 

Although existing data have suggested benefits and drawbacks of alcohol consumption on CVD, to the best of our knowledge, no study has looked into the association between alcohol consumption and cardiovascular health (CVH), a seven-metrics score proposed by the American Heart Association (AHA) to estimate, promote, and monitor CVH at the population level [12,13]. Our hypothesis was that the assessment of CVH could incorporate detailed information on alcohol consumption and drinking patterns. Thus, we investigated the association between alcohol consumption and CVH, with the aim of understanding whether alcohol quantity, drinking patterns, and types of alcoholic beverage might exert different effects on their relationship. For doing that, we conducted a cross-sectional analysis on data from the Kardiovize study, a population-based prospective study assessing both traditional and novel risk factors for CVD, in a randomly selected ~1% of the urban population of Brno (Czech Republic) [14,15,16,17,18,19,20,21,22,23]. We have chosen this cohort since previous findings from the Kardiovize study were consistent with the WHO report in showing that approximately 80% Czech adult drank alcohol in the form of beer, spirits, or wine [1,24]. This places Czech Republic at the top of countries for alcohol consumption—only behind Moldova and Lithuania—with 14.4 L of pure alcohol per adult [1].

## 2. Materials and Methods

### 2.1. Study Design and Participants

The Kardiovize Brno 2030 cohort includes a random sample of residents of working age (aged 25–64 years) of the city of Brno, Czech Republic [17]. The baseline study protocol was approved by the ethics committee of St Anne’s University Hospital, Brno, Czech Republic (reference 2 G/2012), in accordance with the Declaration of Helsinki, and all participants signed an informed consent to participate in the study. All patients were recruited from 2013 to 2014 and data used in the present study were collected at the same time during the baseline visit. Data on demographic and socioeconomic status (age, gender, educational level, employment, and marital status), behaviors (smoking status, diet, and physical activity), and personal history of diseases were collected by face-to-face health comprehensive interviews [17]. Physical activity was evaluated using the long version of the International Physical Activity Questionnaire (IPAQ-L) translated into Czech [18,25]. Dietary assessment was performed by a 43-item Food Frequency Questionnaire (FFQ) using the previous week as the reference period [14,15,26]. During the baseline visit, physical examinations and anthropometric measurements were performed by trained researchers according to standardized techniques and previously described protocols [16,17]. In this cross-sectional analysis, we used data from Kardiovize participants with complete assessment of alcohol consumption, CVH, and no history of CVD. To reduce reporting bias in dietary assessment, we also excluded participants with implausible energy intakes (reported as <600 or >3600 kcal/day for women and <800 or >4200 kcal/day for men).

### 2.2. Alcohol Consumption Assessment

Alcohol consumption was assessed by the reported alcohol intake of the last week, expressed in number of standard drinks. Standard drinks were assessed as a glass of wine, a bottle of beer, or a shot of spirits, which approximately contained 10 g ethanol [27]. Next, we calculated the number of standard drinks per week and classified participants as non-drinkers (including abstainers and those who did not drink in the previous 12 months), light-to-moderate (1–7 or 1–14 standard drinks per week for women and men, respectively), or heavy (>7 or >14 standard drinks per week for women and men, respectively) drinkers [27]. We also assessed the number of drinking days per week. Furthermore, for participants who drank more than one standard drink per week, we defined exclusively drinkers (i.e., drinking one alcoholic beverage type exclusively), those who preferred one alcoholic beverage type (i.e., drinking more frequently beer, wine, or spirits than the other two types), and those without a preference (i.e., drinking two or three types with equal frequency). To evaluate the independent effect of different types of alcoholic beverage on CVH among current drinkers, we also calculated the number of standard drinks per week of wine, beer, and spirits. For each type of alcoholic beverage, drinkers were classified as non-drinkers, light-to-moderate (1–7 or 1–14 standard drinks per week for women and men, respectively), or heavy (>7 or >14 standard drinks per week for women and men, respectively) drinkers [27].

### 2.3. Assessment of Cardiovascular Health

The CVH score includes 7 health behaviors and health factors-namely body mass index (BMI), healthy diet, physical activity, smoking status, blood pressure, blood glucose, and total cholesterol—which can help identify people with poor CVH at higher risk for CVD [12]. Thus, CVH score was computed as the sum of seven components (i.e., BMI, healthy diet, physical activity level, smoking status, blood pressure, blood glucose, and total cholesterol) according to the definition by the AHA [12,13]. Each component was scored from 0–2 (0 = poor, 1 = intermediate, and 2 = ideal) and therefore the overall CVH score ranged from 0 to 14 (Appendix A). Accordingly, CVH status was defined as ideal (all seven components being at ideal levels), intermediate (at least one component at intermediate level, but no poor components), and poor (at least one of seven components at poor level) [28].

### 2.4. Statistical Analyses

The normality of continuous variables was first tested using the Kolmogorov–Smirnov test. Accordingly, variables with skewed distribution were reported as median (interquartile range (IQR)) and compared with the Mann–Whitney U test. The Chi-square test, instead, was used to compare categorical variables, which were reported as frequency (percentage, %). Multiple logistic regression models were employed to assess the association of alcohol consumption with CVH status. Different models worked on alcohol consumption as; (i) number of standard drinks per week, (ii) abstainers vs. drinkers, (iii) categorized into abstainers, light-to-moderate, and heavy drinkers, and (iv) categorized as preponderant or exclusively drinking of each type of alcoholic beverage. In this paper, we presented results as odds ratio (OR) and the corresponding 95%CI of poor CVH using non-drinkers (including abstainers and those who did not drink in the previous 12 months) as reference group. To test the robustness of our findings, we also performed a sensitivity analysis by removing abstainers from the reference group. Among drinkers, we simultaneously modelled the amount of different types of alcoholic beverage (i.e., categorized as non-drinking, light-to-moderate drinking, or heavy drinking) to evaluate their independent effect on poor CVH. We also investigated for interactions in the association of alcohol consumption and poor CVH. Regression models were applied to the whole cohort and stratified by age or gender, adjusting for the number of drinking days and for variables significantly associated with alcohol consumption and not included in the CVH score (i.e., age, gender, educational level, and employment). The SPSS software (version 22.0, SPSS, Chicago, IL, USA) was used in all statistical analyses. All tests were 2-sided, and *p*-values < 0.05 were considered statistically significant.

## 3. Results

### 3.1. Relations of Total Alcohol Intake with Cardiovascular Risk Factors

A total of 1773 Kardiovize participants (aged 25–64 years; 44.2% men) who satisfied the selection criteria were included in the current analysis. Approximately 28.2% of subjects were non-drinkers, while 45.6% and 26.2% were light-to-moderate or heavy drinkers, respectively. In general, drinkers were younger, more likely to be men, more educated, employed, and to smoke tobacco than non-drinkers, with slight differences between light-to-moderate and heavy drinkers (Table 1). With respect to CVD risk factors, drinkers also exhibited higher diastolic blood pressure than non-drinkers (Table 1). While no differences in CVD risk factors were evident when comparing light-to-moderate drinkers with non-drinkers, heavy drinkers were more likely to smoke tobacco, and to display diastolic blood pressure, fasting glucose, triglycerides, and low-density lipoprotein (LDL)-cholesterol at poorer level. By contrast, heavy drinkers were more likely to report high-density lipoprotein (HDL)-cholesterol at good level. 

### 3.2. Total Alcohol Intake and Cardiovascular Health

In general, logistic regression analysis demonstrated that the risk of poor CVH status increased with increasing number of drinks per week (OR = 1.04 per drink per week; 95%CI = 1.02–1.06; *p* < 0.001), while decreased with increasing number of drinking days per week (OR = 0.89; 95%CI = 0.83–0.97; *p =* 0.004). However, the comparison of CVH components across alcohol consumption categories (Figure 1) showed that drinkers, especially heavy drinkers, were more likely to have better levels of physical activity (*p* = 0.035) and weight (*p* < 0.001), and worst level of smoking (*p* = 0.001). However, neither light-to-moderate (OR = 0.87; 95%CI = 0.67–1.13; *p* = 0.290) nor heavy drinkers (OR = 1.29; 95%CI = 0.96–1.73; *p* = 0.086) had significant risk of poor CVH status than non-drinkers. Although findings were not statistically significant, they suggest a potential J-shaped relationship between alcohol intake and poor CVH status (Figure 2).

### 3.3. Total Alcohol Intake and Cardiovascular Health

Regardless of total alcohol intake, we next identified those who prefer, or exclusively drink, a particular type of alcoholic beverage among those who drank more than one standard drink per week. Particularly, exclusively wine drinkers exhibited lower BMI (median = 24.6; IQR = 5.9 vs. median = 25.6; IQR = 7.9; *p* = 0.016), LDL-cholesterol (median = 2.9; IQR = 1.2 vs. median = 3.1; IQR = 1.2; *p* = 0.028), and total cholesterol/HDL-cholesterol ratio (median = 3.1; IQR = 1.2 vs. median = 3.5; IQR = 1.5; *p* < 0.001) than non-drinkers. With respect to CVH metrics, those who preferred to drink wine, or exclusively wine drinkers, were more likely to report weight at intermediate-ideal level (*p* = 0.020 and p<0.001) compared with non-drinkers in general (Figure 3). After adjusting for covariates, however, no association of preponderant or exclusively wine drinking with poor CVH was evident.

By contrast, those who preferred to drink beer, as well as exclusively beer drinkers, exhibited higher systolic (median = 119.0; IQR = 17.3 and median = 120.0; IQR = 21.0 vs. median = 117.0; IQR = 21.5; *p* = 0.003 and *p* = 0.002) and diastolic (median = 80.8; IQR = 10.5 and median = 81.0; IQR = 14.0 vs. median = 78.0; IQR = 13.0; *p* = 0.050 and *p* < 0.001) blood pressure, and triglycerides (median = 1.1; IQR = 0.8 and median = 1.1; IQR = 0.8 vs. median = 1.0; IQR = 0.8; *p* = 0.009 and *p* = 0.002) than non-drinkers. With respect to CVH metrics, those who prefer to drink beer, as well as the exclusively beer drinkers, were less likely to report blood pressure (*p* = 0.001 and *p* < 0.001) and weight (*p* < 0.001 and *p* < 0.001) at ideal level compared with non-drinkers (Figure 3). After adjusting for covariates, however, no association of preponderant or exclusively beer drinking with poor CVH status was evident.

Notably, differences in BMI, systolic and diastolic blood pressure, triglycerides, HDL- and LDL-cholesterol, and total cholesterol/LDL-cholesterol ratio became more significant if comparing exclusively wine with exclusively beer drinkers (data not shown). With respect to CVH metrics, exclusively wine drinkers were more likely to report blood pressure (*p* < 0.001), weight (*p* = 0.002), diet (*p* = 0.007), smoking (*p* < 0.001), and the overall CVH status (*p* = 0.003) at intermediate-ideal level than exclusively beer drinkers (Figure 3). Accordingly, after adjusting for covariates, exclusively wine drinkers showed lower odds of poor CVH status than exclusively beer drinkers (OR = 0.67; 95%CI = 0.48–0.95; *p* = 0.022). Neither preponderant nor exclusively spirit drinkers were associated with CVD risk factors, CVH status, and its components (data not shown).

### 3.4. Relations of Alcohol Consumption with CVH According to Types of Beverages

Among drinkers, we evaluated the effects of the amount of alcohol consumption according to types of alcoholic beverages. We observed that wine drinkers, especially light-to-moderate drinkers, were more likely to report smoking (*p* < 0.001) and have blood pressure (*p* < 0.001) at ideal levels compared with non-wine drinkers. This resulted in lower prevalence of poor CVH status among light-to-moderate wine drinkers (*p* = 0.006) (Appendix A). By contrast, heavy beer drinkers were more likely to report smoking (*p* < 0.001) and have weight (*p* = 0.002) and blood pressure (*p* < 0.001) at poor level, while non-beer drinkers were more likely to report diet at ideal level (*p* = 0.012). This resulted in poorer CVH status among heavy beer drinkers compared with non-beer drinkers (*p* = 0.015) (Appendix A). With respect to spirits, no significant differences in CVH components were evident across categories of consumption (Appendix A). Finally, we simultaneously modelled the amount of different types of alcoholic beverage on poor CVH status using a logistic regression analysis (Figure 4). Notably, we found that heavy beer drinkers had a higher risk of poor CVH status than non-beer drinkers (OR = 1.70; 95%CI = 1.04–2.77; *p* = 0.035), while light-to-moderate wine drinkers had a lower risk of poor CVH status than non-wine drinkers (OR = 0.73; 95%CI = 0.54–0.99; *p* = 0.045).

### 3.5. Covariate Relations

We found significant interactions between alcohol consumption, age, and gender (*p*-values < 0.01), but not with educational level and employment status (*p* = 0.378 and *p* = 0.875). Accordingly, we performed logistic regression analyses stratified by age and gender and adjusted for educational level and employment. 

In women aged 25–44 years, the risk of poor CVH status increased with increasing number of drinks per week (OR = 1.07; 95%CI = 1.02–1.16; *p* = 0.003; for 1 unit increase in the number of standard drinks per week). Although light-to-moderate drinking did protect from poor CVH status in general, among drinkers, light-to-moderate beer, or wine drinkers had a decreased risk of poor CVH status (OR = 0.39; 95%CI = 0.18–0.84; *p* = 0.016; OR = 0.27; 95%CI = 0.11–0.64; *p* = 0.003; respectively). By contrast, heavy drinkers had an increased risk of poor CVH status (OR = 2.43; 95%CI = 1.36–4.34; *p* = 0.003) with no differences between types of beverages. 

Women aged 45–65 years who drank had a decreased risk of poor CVH status than non-drinkers (OR = 0.67; 95%CI = 0.45–0.99; *p* = 0.047). Particularly, this evidence appeared to be attributed to benefits from light-to-moderate drinking (OR = 0.60; 95%CI = 0.39–0.91; *p* = 0.017). Contrary to young women, no association of specific alcoholic beverages with CVH status was evident. Interestingly, no significant association between amount of alcohol consumption, its frequency, or types of beverages and poor CVH status was evident in men across all age ranges (25–65 years). Sensitivity analysis confirmed the robustness of our findings by excluding abstainers from the reference group (data not shown).

## 4. Discussion

In a large prospective cohort in Czech Republic, the Kardiovize study, we conducted a cross-sectional analysis to estimate the association of alcohol consumption with CVH status defined by the AHA. Overall, we observed that drinkers were more likely to have the recommended level of physical activity and weight status, but they were also more likely to smoke. However, we also reported several differences in cardiovascular parameters, most of them not favorable for heavy drinkers. Although our findings did not demonstrate significant associations of light-to-moderate or heavy drinking with the risk of poor CVH status, they suggested a potential J-shaped relationship. In fact, compared with non-drinkers, the risk of poor CVH seemed lower among light-to-moderate drinkers and higher among heavy drinkers. This is in line with the fact that the effects of alcohol are related to the amount consumed and on drinking patterns. In general, most epidemiological studies have reported that light-to-moderate drinkers had less risk of mortality and CVD than abstainers, while heavy drinkers were at the highest risk [29]. In line, others reported benefits of light-to-moderate drinking on CVD risk reduction, including congestive heart failure [30], ischemic stroke [31,32], atherosclerosis [33,34,35], peripheral arterial disease [36], hemorrhagic stroke [37], and myocardial infarction [38,39]. Particularly, the INTER-HEART study found that daily but moderate alcohol consumption was associated with a reduced risk of myocardial infarction, independent of age, sex, and history of diseases [38]. Researchers of the Health Professionals Follow-Up Study, moreover, demonstrated a positive effect of regular drinking on myocardial infarction risk among men following a very healthy lifestyle [39]. In our study, better scores of physical activity and weight status among drinkers probably attenuated the association between alcohol consumption and poor CVH.

Some of the positive effects of drinking on CVDs support the so-called “French Paradox”, which describes the inverse relationship between alcohol consumption and CVD observed in France, mostly attributed to wine [40]. Thus, we evaluated how beverage choices could differently affect CVD risk factors and CVH status. Notably, the comparison with non-drinkers demonstrated that exclusively wine drinkers had lower BMI, LDL-cholesterol, and cholesterol ratio, being more prone to achieve the weight status recommended by the AHA. Cardiovascular benefits of wine were also confirmed among drinkers: Compared with non-wine drinkers, in fact, light-to-moderate wine drinkers reported better status for smoking and blood pressure, which resulted in a lower prevalence of poor CVH status. This is in line with several epidemiological studies in favor of wine conferring cardiovascular benefits. One of the largest studies on the health effects of alcoholic beverage choice was conducted by Gronbaek and colleagues among about 13,000 Danes: Daily moderate wine drinking significantly reduced the risk of cardiovascular death compared to non-drinking [41]. This evidence was further confirmed by other prospective studies [40,42], showing lower cardiovascular mortality in moderate wine drinkers than in moderate beer drinkers or spirit drinkers.

In our study, beer drinkers were less likely to achieve blood pressure and weight status recommended by the CVH metrics compared with non-beer drinkers. Moreover, differences in CVH metrics between heavy beer drinkers and non-beer drinkers resulted in higher prevalence of poor CVH status among the former ones. Benefits and drawbacks of beverage choice became more significant if comparing exclusively wine with exclusively beer drinkers. In line, exclusively wine drinkers were more likely to achieve the CVH recommendation of ideal blood pressure, weight, diet, and smoking, which resulted in a better CVH. Particularly, exclusively wine drinkers showed lower odds of poor CVH status than exclusively beer drinkers, independent of age, sex, educational level, employment status, and number of drinking days per week. 

Cardiovascular benefits of wine drinking are generally attributed to flavonoids and nonflavonoids that compose wine [43,44,45,46,47]. However, we showed that wine drinkers can be viewed as healthy beyond cardiovascular parameters. Indeed, it seemed reasonable that they also payed more attention to their behaviors. In light of these considerations, some questions about current knowledge arose. Leaving aside biological explanations in support of wine that we already know, is it possible that part of beneficial effect of drinking wine against CVDs has been due to incomplete adjustment for other behaviors? And again, can we consider drinking wine as an indicator of more attention to personal CVH? Our findings, at least partly, supported that drinking habits could be included in the assessment of CVH status, because of their associations with behavioral and clinical items. Anyway, it is obvious that direct benefits of drinking wine through the inhibition of LDL oxidation [46], anti-atherosclerotic, and antioxidant properties [47] should not be neglected.

Strengths of our study included a detailed assessment of alcohol consumption, in terms of quantity, frequency, and beverage choices. Moreover, our findings might be generalized to the Czech population, since the Kardiovize study was a population-based cohort, which randomly selected a sample of the urban population of Brno (Czech Republic). Our study used comprehensive health interviews and examinations which were performed using standardized and validated protocols, which allowed to use a composite measure of CVH. The large number of participants and the availability of information also allowed us to examine association by age and gender and to perform sensitivity analysis by removing abstainers from the analyses. However, our study had also some limitations. First, data on alcohol consumption were self-reported, which did not preclude potential measurement errors and may suffer from inaccuracies. Moreover, information on wine drinking did not distinguish between red and white wine. Second, we had limited information to evaluate the influence of binge drinking. Third, the cross-sectional design did not allow us to assess for causality. Finally, some protective or harmful effects of alcohol consumption could be attributed to social factors [48,49,50]. In our study, for instance, we confirmed that some effects of alcohol might be influenced by age and sex. In fact, stratified analyses in men failed in demonstrating a significant association of alcohol consumption with CVH status, considering quantity, frequency, and alcoholic beverage preference. In young women, instead, we observed that the risk of poor CVH status was the highest among heavy drinkers and the lowest among low-to-moderate drinkers. Despite these adjustments, however, we cannot completely exclude the effect of other unmeasured factors. Finally, it is worth mentioning that CVH metrics refer to behavioral and clinical risk factors assessed among individuals with no history of CVDs. For this reason, our findings should be considered with a view to benefits and drawbacks for cardiovascular outcomes that could arise in the future. Based on these considerations, further large-size prospective studies should be encouraged to confirm our findings in other populations and to evaluate the long-term effect of drinking alcohol and beverage choices on cardiovascular outcomes.

## 5. Conclusions

In summary, our study provided a comprehensive assessment of the association of alcohol consumption—expressed as quantity, frequency, and drinking patterns—with CVH status. We found that this relationship was related to the amount of alcohol consumed, drinking patterns, and beverage choices. Noteworthy, we pointed out that drinking habits were associated with both behaviors and cardiovascular parameters. Although these findings supported the hypothesis that drinking alcohol was related to the CVH in general, further prospective research is needed to understand whether the assessment of CVH status should incorporate information on alcohol consumption.

## Figures and Tables

**Figure 1 nutrients-12-02848-f001:**
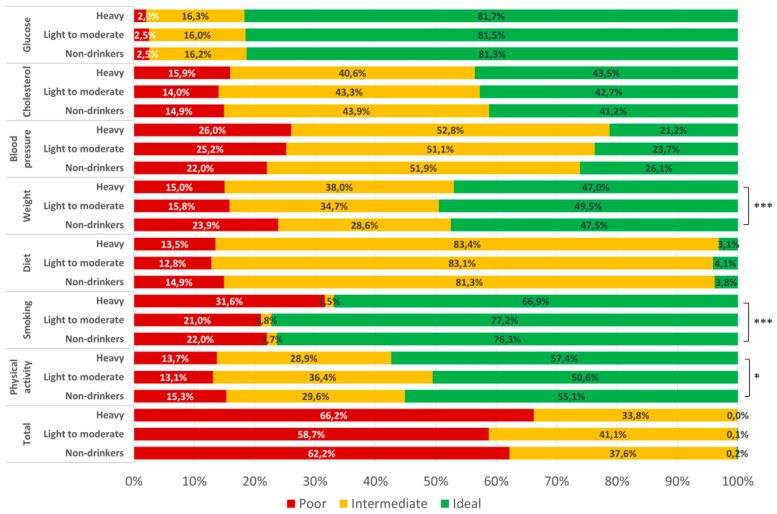
Distribution of cardiovascular health metrics according to drinking patterns. * *p* < 0.05, *** *p* < 0.001 based on the Chi-Squared test using non-drinkers as reference group.

**Figure 2 nutrients-12-02848-f002:**
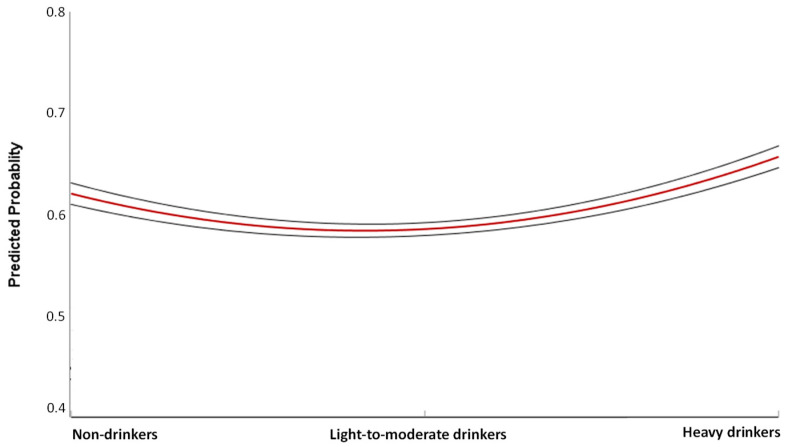
Relationship between categories of alcohol consumption and poor cardiovascular health status. This graph shows the predicted probability (red line) and 95% confidence intervals (grey lines) of poor cardiovascular health status according to categories of alcohol consumption.

**Figure 3 nutrients-12-02848-f003:**
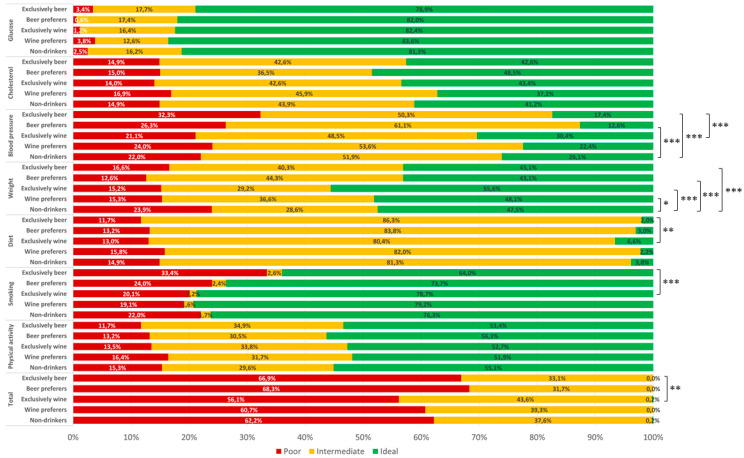
Distribution of cardiovascular health metrics according to beverage choices. * *p* < 0.05, ** *p* < 0.01, *** *p* < 0.001 based on the Chi-Squared test using non-drinkers as reference group.

**Figure 4 nutrients-12-02848-f004:**
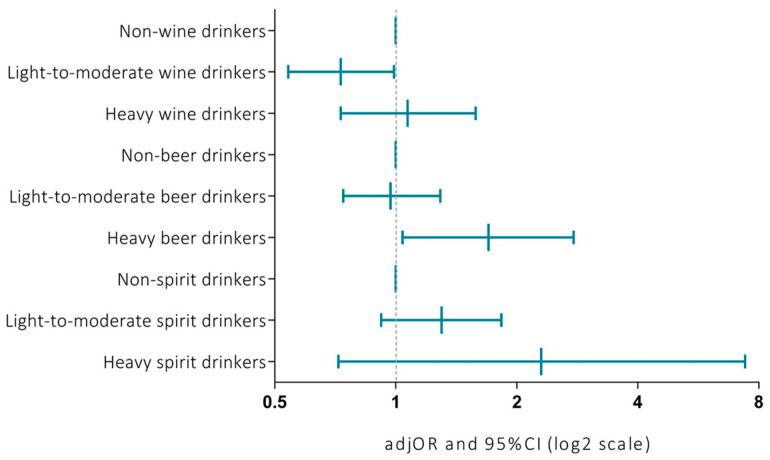
Association of alcohol consumption with poor cardiovascular health status according to beverage choices. This graph shows adjusted odds ratios (ORs) and 95CI% intervals of the association between categories of alcohol consumption and poor cardiovascular health status. Results are based on logistic regression models and adjusted for number of drinking days, age, gender, educational level, and employment status. Non-drinkers of each specific alcoholic beverage were used as reference groups.

**Table 1 nutrients-12-02848-t001:** Comparison of socio-demographic characteristics and cardiovascular diseases (CVD) risk factors between drinkers and non-drinkers.

Characteristics	Non-Drinkers (*n* = 500)	Drinkers					
		Light to Moderate (*n* = 808)		Heavy (*n* = 465)		Overall (*n* = 1273)	
	Median (IQR) or %	Median (IQR) or %	*p-*Value	Median (IQR) or %	*p*-Value	Median (IQR) or %	*p*-Value
Age, years	49.0 (19.0)	47.0 (19.0)	0.310	45.0 (19.0)	0.001	46.0 (19.0)	0.028
Sex (% male)	27.3%	50.4%	<0.001	51.6%	<0.001	50.9%	<0.001
Educational level (% low) *	25.4%	16.7%	<0.001	15.5%	<0.001	16.3%	<0.001
Marital status (% living alone)	39.5%	35.5%	0.147	41.5%	0.538	37.7%	0.477
Employment (% unemployed)	25.1%	16.8%	0.001	13.6%	<0.001	15.6%	<0.001
Smoking (% current smokers)	24.9%	23.8%	0.757	34.1%	<0.001	27.6%	0.077
BMI, kg/m^2^	25.6 (7.8)	25.1 (5.9)	0.277	25.4 (5.6)	0.548	25.2 (7.8)	0.312
Waist circumference, cm	88.0 (22.0)	88.0 (20.0)	0.430	89.0 (19.0)	0.270	88.0 (20.0)	0.302
Systolic Blood Pressure, mmHg	117.0 (21.5)	118.0 (18.5)	0.590	119.0 (19.0)	0.083	118.5 (18.5)	0.254
Diastolic Blood Pressure, mmHg	78.0 (12.5)	78.5 (12.5)	0.072	80.5 (13.0)	<0.001	79.5 (12.5)	0.002
Diagnosis of Hypertension	30.9%	31.1%	0.921	29.3%	0.610	30.5%	0.874
Fasting Glucose, mmol/L	4.9 (0.7)	4.9 (0.7)	0.387	5.0 (0.7)	0.021	4.9 (0.7)	0.103
Diagnosis of Diabetes	10.4%	9.0%	0.415	6.8%	0.057	8.2%	0.147
Triglycerides, mmol/L	1.03 (0.90)	1.03 (0.80)	0.609	1.08 (0.80)	0.022	1.05 (0.80)	0.167
Total Cholesterol, mmol/L	5.13 (1.40)	5.11 (1.30)	0.784	5.12 (1.20)	0.398	5.11 (1.30)	0.571
HDL Cholesterol, mmol/L	1.47 (0.50)	1.47 (0.50)	0.624	1.56 (0.50)	0.001	1.50 (0.50)	0.061
LDL Cholesterol, mmol/L	3.14 (1.30)	2.97 (1.20)	0.641	3.03 (1.2)	0.035	3.00 (1.20)	0.206
Cholesterol/HDL-Cholesterol ratio	3.46 (1.50)	3.27 (1.40)	0.549	3.46 (1.50)	0.005	3.38 (1.40)	0.094
Diagnosis of hypercholesterolemia	28.7%	29.3%	0.817	25.7%	0.299	28.0%	0.766
Cardiovascular Health Score †	9.0 (3.0)	9.0 (3.0)	0.488	9.0 (3.0)	0.476	9.0 (3.0)	0.880

Results are reported as median (interquartile range (IQR)) or percentage. Statistical analyses were conducted by Chi-square test for bivariate or categorical variable, and Mann–Whitney U test for continuous variables using non-drinkers as reference group. * Primary education or apprenticeship. † Computed as the sum of seven metrics defined by the American Heart Association (AHA). Abbreviations: IQR, interquartile range; BMI, body mass index; HDL, high-density lipoprotein; and LDL, low-density lipoprotein.

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
