# Peer review of "Is Drinking Alcohol Really Linked to Cardiovascular Health? Evidence from the Kardiovize 2030 Project"

_nutrients, 2020, doi:10.3390/nu12092848_

Round 1

Reviewer 1 Report

The current study uses data from the Kardiovize 2030 project to understand the relationship between cardiovascular health and alcohol consumption. Data from individuals regarding cardiovascular health is stratified according to a validated AHA scoring system and alcohol consumption is self-reported. The study finds differences in individuals who were wine drinkers – associated with better cardiovascular health score – than beer drinkers.

  1. The study concept is novel and interesting. The study, however, doesn’t have information on cardiovascular outcomes so much of what is stated needs to be toned down and this has to be acknowledged as a limitation. Other things to consider:
  2. What time frame was the study data collected and was the cardiovascular risk factor data collected at the same time as the alcohol intake data?
  3. Did your alcohol intake wine category collect information on red vs. white wine? This is important for cardiovascular benefits of wine arguments. If not, it’s a limitation.
  4. Were all measurements – clinical, BP, blood tests, etc. done at the same time as the survey questions? You refer to some of these variables as patient-reported.
  5. In your tables, please put the number of individuals per category.
  6. What reference did you use to create the alcohol intake categories?
  7. Fig. 1 – you have a discordance between weight and physical activity with respect to categories. Is there an explanation?
  8. Throughout the paper you refer to poor cardiovascular health and protection from poor cardiovascular health; however, you are only relating alcohol consumption to a score and you don’t have outcomes. This means that you need to change your comments to poor CVH score. What constitutes a poor vs an intermediate vs ideal overall score? Presumably 0 is poor, 7 is intermediate, and 14 is ideal based on how you use the AHA score but what about in-between scores?
  9. Also need to state that this study was limited to a single population and needs to be confirmed in other populations.

Author Response

Dear Editor,

this document is intended for the convenience of the editor and reviewers and contains the list of the requested changes. We would like to take this opportunity to thank the reviewers for the insightful comments and suggestions to improve the paper. We hereby submit to your attention a revised version of the manuscript in which we have considered all comments. The following list of changes and answers to comments of Reviewers addresses all changes made in the manuscript (in red font).

Comments submitted by Reviewer 1

The current study uses data from the Kardiovize 2030 project to understand the relationship between cardiovascular health and alcohol consumption. Data from individuals regarding cardiovascular health is stratified according to a validated AHA scoring system and alcohol consumption is self-reported. The study finds differences in individuals who were wine drinkers – associated with better cardiovascular health score – than beer drinkers.

Answer: We are very grateful with Reviewer 1 for his/her positive comments on our manuscript.

  1. The study concept is novel and interesting. The study, however, doesn’t have information on cardiovascular outcomes so much of what is stated needs to be toned down and this has to be acknowledged as a limitation. Other things to consider:

 Answer: We agree with this comment by Reviewer 1 and thus we commented this point in the discussion section (lines 324-330)

  1. What time frame was the study data collected and was the cardiovascular risk factor data collected at the same time as the alcohol intake data?

Answer: We apologize if the time frame of our study was not indicated in the previous version of our manuscript. According to this comment, we explained that all patients were recruited from 2013 to 2014 and that all the data used in this study were collected a the same time during the baseline visit (lines 78-79).

  1. Did your alcohol intake wine category collect information on red vs. white wine? This is important for cardiovascular benefits of wine arguments. If not, it’s a limitation.

Answer: This is an important consideration that underlines a limitation of our study. Accordingly, we explained in the discussion section that wine category did not distinguish between red and white wine (lines 315-316).

  1. Were all measurements – clinical, BP, blood tests, etc. done at the same time as the survey questions? You refer to some of these variables as patient-reported.

Answer: As described in the method section (lines 85-87), physical examinations and anthropometric measurements were performed by trained researchers according to standardized techniques and previously described protocols.

  1. In your tables, please put the number of individuals per category

Answer: As suggested, we included the number of individuals in table 1. .

  1. What reference did you use to create the alcohol intake categories?

Answer: We are sorry if reference used for alcohol assessment was not clear. In the revised version of our manuscript we better indicated that alcohol categories were defined according to reference 28.

  1. Fig. 1 – you have a discordance between weight and physical activity with respect to categories. Is there an explanation?

Answer: As reported in the Supplementary Table 1, the definition of CVH metrics for weight and physical activity were different and no apparent association is evident. However, it is worth mentioning that Figure 1 depicted the bivariate analysis between CVH metrics and drinking patterns and not the multivariable effect of drinking alcohol on CVH. Differences in terms of age, sex or other factors might partially explain this controversy.

  1. Throughout the paper you refer to poor cardiovascular health and protection from poor cardiovascular health; however, you are only relating alcohol consumption to a score and you don’t have outcomes. This means that you need to change your comments to poor CVH score. What constitutes a poor vs an intermediate vs ideal overall score? Presumably 0 is poor, 7 is intermediate, and 14 is ideal based on how you use the AHA score but what about in-between scores?

Answer: We agree with this comment and thus we changed “poor CVH” with “poor CVH status”. The definition of poor CVH status is reported in lines 116-119.

  1. Also need to state that this study was limited to a single population and needs to be confirmed in other populations.

Answer: As suggested by Reviewer 1, we included an encouragement to further research in order to confirm our findings in other populations (lines 328-331).

Reviewer 2 Report

Summary:

The present cross-sectional study analyzed the association between alcohol consumption and Cardiovascular Diseases (CVD) score proposed by the American Heart Association. This study was mainly focused on the comparison of CVD risk factors and CVH metrics between non-drinkers vs. drinkers (light to moderate and heavy), and within the drinkers between light to moderate vs. heavy drinkers, between different types of alcohol consumers (exclusive vs. preferred), and also compared between genders (male vs female) and at different aged people within the gender. The authors reported that light to moderate alcohol drinkers had lower poor CVH metrics whereas heavy drinkers had higher poor CVH metrics compared to the non-drinkers. In addition, people who drink exclusively wine exhibited better CVH than those drink exclusively beer. Light to moderate women drinkers aged between 25-44 yrs. protected from poor CVH than older (45-65 yrs.) and no significant association was found in all men (25-65 yrs.). Overall, this study enhanced the understanding of association between alcoholic consumption and risk of cardiovascular diseases.

Here are my suggestions.

  • Smoking tobacco increases the risk of CVDs and the comparison of CVD risk factors and cardiovascular health (CVH) score between heavy drinkers with and without smoking habits and with non-drinkers is missing?
  • Need to double check the statistical analysis of fasting blood glucose levels, diastolic blood pressure, triglyceride levels, HDL and LDL data between non-drinkers vs. drinkers. It looks like the difference between the groups was minimal and interquartile range (median) was higher within the group in all groups.

Author Response

Dear Editor,

this document is intended for the convenience of the editor and reviewers and contains the list of the requested changes. We would like to take this opportunity to thank the reviewers for the insightful comments and suggestions to improve the paper. We hereby submit to your attention a revised version of the manuscript in which we have considered all comments. The following list of changes and answers to comments of Reviewers addresses all changes made in the manuscript (in red font).

Comments submitted by Reviewer 2

Summary:

The present cross-sectional study analyzed the association between alcohol consumption and Cardiovascular Diseases (CVD) score proposed by the American Heart Association. This study was mainly focused on the comparison of CVD risk factors and CVH metrics between non-drinkers vs. drinkers (light to moderate and heavy), and within the drinkers between light to moderate vs. heavy drinkers, between different types of alcohol consumers (exclusive vs. preferred), and also compared between genders (male vs female) and at different aged people within the gender. The authors reported that light to moderate alcohol drinkers had lower poor CVH metrics whereas heavy drinkers had higher poor CVH metrics compared to the non-drinkers. In addition, people who drink exclusively wine exhibited better CVH than those drink exclusively beer. Light to moderate women drinkers aged between 25-44 yrs. protected from poor CVH than older (45-65 yrs.) and no significant association was found in all men (25-65 yrs.). Overall, this study enhanced the understanding of association between alcoholic consumption and risk of cardiovascular diseases.

Answer: We are very grateful with Reviewer 2 for his/her positive comments on our manuscript.

Here are my suggestions.

  • Smoking tobacco increases the risk of CVDs and the comparison of CVD risk factors and cardiovascular health (CVH) score between heavy drinkers with and without smoking habits and with non-drinkers is missing?

Answer: We agree with Reviewer 2 that smoking status might interact with drinking habits on CVDs. However, our study aims to assess the association between drinking alcohol and CVH, a score proposed by the AHA to evaluate the risk among individuals with no history of CVDs. Moreover, smoking status is a metric included in the CVH assessment and thus it is not possible to evaluate the association between drinking habits and CVH stratifying by smoking status.

  • Need to double check the statistical analysis of fasting blood glucose levels, diastolic blood pressure, triglyceride levels, HDL and LDL data between non-drinkers vs. drinkers. It looks like the difference between the groups was minimal and interquartile range (median) was higher within the group in all groups.

Answer: Table 1 reports the comparison of behavioral and clinical risk factors between non-drinkers and drinkers in terms of median (IQR). As indicated in the table, no statistically significant differences were evident between these groups except for diastolic blood pressure. The double-checking of this analysis confirmed the results reported in Table 1. The high variation within each group (as indicated by the IQR) might be partially attributable to the heterogeneity in the study population.